# *phox2ba:* The Potential Genetic Link behind the Overlap in the Symptomatology between CHARGE and Central Congenital Hypoventilation Syndromes

**DOI:** 10.3390/genes14051086

**Published:** 2023-05-15

**Authors:** Jessica E. MacLean, Jaime N. Wertman, Sergey V. Prykhozhij, Emily Chedrawe, Stewart Langley, Shelby L. Steele, Kevin Ban, Kim Blake, Jason N. Berman

**Affiliations:** 1Department of Pediatrics, Dalhousie University, Halifax, NS B3K 6R8, Canada; 2Department of Microbiology and Immunology, Dalhousie University, Halifax, NS B3H 4R2, Canada; 3Children’s Hospital of Eastern Ontario Research Institute, Ottawa, ON K1H 8L1, Canada; 4Departments of Pediatrics and Cellular and Molecular Medicine, University of Ottawa, Ottawa, ON K1N 6N5, Canada

**Keywords:** CHARGE syndrome, zebrafish, CHD7, anesthesia, PHOX2B

## Abstract

CHARGE syndrome typically results from mutations in the gene encoding chromodomain helicase DNA-binding protein 7 (*CHD7*). CHD7 is involved in regulating neural crest development, which gives rise to tissues of the skull/face and the autonomic nervous system (ANS). Individuals with CHARGE syndrome are frequently born with anomalies requiring multiple surgeries and often experience adverse events post-anesthesia, including oxygen desaturations, decreased respiratory rates, and heart rate abnormalities. Central congenital hypoventilation syndrome (CCHS) affects ANS components that regulate breathing. Its hallmark feature is hypoventilation during sleep, clinically resembling observations in anesthetized CHARGE patients. Loss of *PHOX2B* (paired-like homeobox 2b) underlies CCHS. Employing a *chd7*-null zebrafish model, we investigated physiologic responses to anesthesia and compared these to loss of *phox2b*. Heart rates were lower in *chd7* mutants compared to the wild-type. Exposure to tricaine, a zebrafish anesthetic/muscle relaxant, revealed that *chd7* mutants took longer to become anesthetized, with higher respiratory rates during recovery. *chd7* mutant larvae demonstrated unique *phox2ba* expression patterns. The knockdown of *phox2ba* reduced larval heart rates similar to *chd7* mutants. *chd7* mutant fish are a valuable preclinical model to investigate anesthesia in CHARGE syndrome and reveal a novel functional link between CHARGE syndrome and CCHS.

## 1. Introduction

CHARGE is an acronym for ocular coloboma, heart defects, choanal atresia, retardation of growth and development, genitourinary malformation, and ear abnormalities, all of which are common clinical features of CHARGE syndrome [1]. CHARGE syndrome is estimated to occur in 1/10,000 live births [2]. Mutations in the chromodomain helicase DNA binding protein 7 gene (*CHD7*) [3] occur in 80% of individuals with the syndrome [2,4]. CHD7 is involved in the regulation of neural crest development, a process during embryogenesis giving rise to tissues of the face and skull and forming components of the autonomic nervous system (ANS), which is responsible for modulating the respiratory, cardiovascular, and digestive systems [2,5]. The multiple anomalies, including choanal atresia, tracheoesophageal fistula, and cardiovascular malformations that frequently manifest in CHARGE syndrome, often necessitate surgical repair, requiring anesthesia. CHARGE syndrome patients frequently experience adverse reactions to anesthesia, including heart rate abnormalities, oxygen desaturations, and decreased respiratory rates, which are particularly disconcerting considering the number of procedures under an anesthetic that these individuals often experience [6]. It has been reported that CHARGE patients require higher doses of medication to become anesthetized, suggesting they respond atypically to anesthesia [7].

The paired-like homeobox 2B (*PHOX2B*) gene is also involved in neural crest development, and loss of function mutations result in Central Congenital Hypoventilation Syndrome (CCHS). CCHS is characterized by dysregulation of the ANS, resulting in the failure of the respiratory system to sense increasing CO_2_ levels, usually during sleep [5,8,9].

CHARGE syndrome and CCHS share several features. They are both caused by mutations in single genes involved in neural crest development and impair ANS regulation. Like patients with CHARGE syndrome, individuals with CCHS are also at an increased risk of adverse events following anesthesia [10] These clinical similarities and shared developmental origins warrant further investigation into a possible association between *CHD7* and *PHOX2B* and a potential role for *PHOX2B* in CHARGE syndrome.

Zebrafish are well-recognized as an excellent model for the study of vertebrate development and to model human developmental disorders [11,12]. The larvae are naturally optically transparent, offer inexpensive husbandry, reproduce rapidly, and have many anatomical and physiological features conserved with mammals [13]. Important for the study of CHARGE syndrome, zebrafish neural crest development is highly conserved and well-characterized [14,15]. The zebrafish has served as a model organism for several studies investigating CHARGE syndrome [16,17]. Our group previously employed CRISPR/Cas9 genome editing technology to create a loss of function frameshift mutation in the *chd7* gene, generating a zebrafish model of CHARGE [18], which recapitulates many of the features of the human syndrome [19]. 

We aimed to investigate the expression of *phox2ba* and *phox2bb* genes and anesthetic recovery behavior in a zebrafish model of CHARGE syndrome to further understand the causative factors of the adverse events experienced by CHARGE patients undergoing anesthesia and sedation. We observed lower heart rates in our *chd7* mutant fish and a difference in time to reach anesthesia, with higher respiratory rates during anesthetic recovery. Knockdown of *phox2ba* also reduced heart rates in zebrafish larvae. *chd7* mutant larvae also had lower expression of *phox2ba*.

## 2. Materials and Methods

### 2.1. Zebrafish Husbandry and Genetic Lines

All zebrafish were housed and maintained in a recirculating commercial housing system (Pentair, Apopka, FL, USA) at 28 °C in 14 h:10 h light:dark conditions and bred according to standard protocol [20]. Embryos were collected and grown in E3 medium (5 mM NaCl, 0.17 mM KCl, 0.33 mM CaCl_2_, 0.33 mM MgSO_4_) in incubators maintained at 28 °C. Embryo plates were cleaned every 24 h, removing debris, dead embryos, and chorion material. E3 media was also changed every 24 h. All larvae were used experimentally before 7 days post-fertilization (dpf) unless otherwise noted. The use of zebrafish in this study was approved by the Dalhousie University Committee on Laboratory Animals under protocols #17-131, #17-055, and #17-141. When required for the experiment, embryos and larvae were euthanized by exposure to 10% tricaine solution (0.4%) prepared in E3 media and observed under a dissecting microscope until the heart stopped beating. They were then discarded, as outlined and approved by the Dalhousie University Committee on Laboratory Animals. 

A zebrafish model of CHARGE syndrome was created in a Tübingen wild-type (WT) strain background using clustered regularly interspaced short palindromic repeats (CRISPR)/Cas9-mediated mutation of *chd7* in our lab, as described previously [18,19]. This resulted in a 2-base pair deletion, resulting in a loss-of-function frameshift mutation in the zebrafish *chd7* gene. These *chd7* mutant larvae (both *chd7*^+/−^ and *chd7*^−*/*−^) exhibit similar developmental abnormalities to those seen in *chd7* morphants, including pericardial edema, curvature of the spine, small eyes, and cardiomegaly [19]. WT Tübingen, Casper, and genotyped *chd7^+/+^* animals served as controls in these experiments. 

### 2.2. Genotyping Mutant Lines

Genotyping of the *chd7* CRISPR fish was originally done using T7 Endonuclease I (M0302S; New England Biolabs Inc., Ipswich, MA, USA) digestion according to the manufacturer’s instructions. Subsequently, genotyping was completed using a modified heteroduplex mobility assay (HMA) similar to previously published protocols [21]. Larvae were sacrificed, and genomic DNA was extracted using NaOH. For both assays, we used *chd7* primers with the following sequences: forward -AGG ACA TGC CAT TCA CTG GT, reverse—GAG CAC ACA TTC CCT GTC CT, producing a 369 bp amplicon. PCR was performed using Taq polymerase (10342020; Invitrogen, Waltham, MA, USA), and the resulting fragment was run on an 8% polyacrylamide gel. An alternative genotyping procedure relied on digestion of the same *chd7* PCR reactions by SmlI (M0302S; New England Biolabs Inc., Ipswich, MA, USA) (7 µL of PCR reaction per 20 µL digestion reaction) with wild-type PCR products being completely digestible, heterozygote products being partially resistant and mutant products being completely resistant.

### 2.3. In Situ Hybridization and Correlation with Genotype

For in situ hybridization, embryos were sacrificed at the indicated time point and fixed in 4% paraformaldehyde overnight at 4 °C. Larvae were then moved to 100% methanol, permeabilized with 10 μg/mL Proteinase K (Roche, Indianapolis, Indiana) for 10–45 min, rinsed in 1X PBS-T and refixed in 4% paraformaldehyde for 20 min. Whole-mount RNA in situ hybridization (WISH) assays were adapted from protocols published previously [22], using digoxigenin (DIG)-labeled RNA probes and BCIP/NBT detection (Roche Diagnostics, Indianapolis, IN, USA). Probes generated for *phox2b* were synthesized from cDNA according to a previously published report [23]. Due to an evolutionary full genome duplication event, there are two homologs of *phox2b* in zebrafish—*phox2ba* and *phox2bb.* Probes were generated against both homologs. To examine the expression of these genes, WISH was performed on the whole progeny resulting from an incross of *chd7^+/-^* fish. Embryo expression patterns of *phox2ba* were then scored into ‘normal’, ‘high’, or ‘low’ expression categories by three independent blinded researchers. These WISH expression patterns were compared to those seen in WT Tübingen controls. Since genotyping requires a significant portion of larval tissue in a separate set of larvae, the fish were euthanized and split transversely with a clean scalpel blade just posterior to the yolk sac. The cephalic halves were fixed for in situ hybridization, and the caudal halves were processed for genotyping to correlate expression levels of *phox2ba* in the caudal half with the *chd7* genotype. Care was taken to ensure the head and tail of each fish were labeled to ensure consistent correlations.

### 2.4. Morpholino Injection

Embryos were obtained by breeding adult Casper [24] zebrafish. Casper zebrafish are wholly optically transparent and were used in this project to enhance visualization of the larval heart. Embryos were injected at the one-cell stage under a ZEISS SteREO Discovery.V8 microscope (Carl Zeiss AG, Oberkochen, Germany) with approximately 1 nL of morpholino (MO) solution using a PLI-100 picoinjector (Harvard Apparatus, Holliston, MA, USA). The solution contained either *phox2ba* MO (GeneTools, AGCCATAAGATTAGAATGCACTGTT), *phox2bb* MO (GeneTools, ATACATTGAAAAGGCTCAGTGGAGA), or mismatch control MO (GeneTools) at the concentrations indicated in the results section, together with 0.05% phenol red for visualization in deionized water. Following injection, embryos were maintained in normal E3 media at 28 °C until the experimental time point. 

### 2.5. Cloning and Expression of phox2ba-sfGFP Fusion

We first generated cDNA using LunaScript RT Supermix kit (E3010S; New England Biolabs Inc., Ipswich, MA, USA) from 28 hpf wild-type embryo RNA extracted using PureLink RNA Mini Kit (12183018A; Thermo Fisher Scientific, Waltham, MA, USA). The *phox2ba* coding sequence was amplified using *phox2ba_BamHI_ATG_for* (ATGCGGATCCAGAAACATGGCTTATGAACGAGGCGT) and *phox2ba_BamHI_linker_rev* (GATCGGATCCACTGCCTCCACCGCCACACAGGCACACAGAATCAA) from cDNA using Q5 High-Fidelity 2X Master Mix (M0492S; New England Biolabs Inc., Ipswich, MA, USA) using the standard Q5 protocol with 65 °C annealing temperature. The resulting PCR product was digested with BamHI enzyme and cloned into the BamHI site of the pCS2+MCS-P2A-sfGFP plasmid (Addgene, 74668). The resulting pCS2+phox2ba-sfGFP plasmid and pCS2+TagRFP were linearized by digestion with NotI-HF and extracted with phenol-chloroform. mRNA synthesis was performed using the mMESSAGE mMACHINE SP6 Transcription Kit (AM1340; Thermo Fisher Scientific, Waltham, MA, USA), and RNAs were purified using LiCl precipitation according to the manufacturer’s instructions. The *phox2ba-sfGFP* and *TagRFP* mRNAs were diluted to 70 and 30 ng/µL, respectively, and 2 nL boli were injected into one-cell stage zebrafish eggs by standard microinjection procedures. Injected embryos were then imaged using the ZEISS Axio Zoom.V16 microscope.

### 2.6. Selection of Anesthetic Concentration

Adult *casper* zebrafish (approx. 3 months of age) were exposed to 1.25, 2.5, or 5% of the tricaine stock solution (0.4%) and observed for up to 8 min to determine the experimental concentration of tricaine that effectively anesthetized fish without being fatal. Zebrafish were not fully anesthetized at 1.25%, and death was observed in several fish at 5% tricaine; therefore, 2.5% tricaine was selected as the optimal concentration for this study.

### 2.7. Opercular Beat Analysis

Adult zebrafish (approx. 3 months of age, *chd7^+/+^*, *chd7^+/^*^−^, and *chd7*^−*/*−^ genotypes) were transferred individually to an experimental arena containing 2.5% tricaine. The tank was recorded using an apparatus that suspended an iPhone over the tank so that the entire tank was within view. The video was recorded in real-time, allowing analysis to occur later. Two key metrics were evaluated following the transfer of fish to the tricaine tank: (1) the length of time it took the fish to lose touch response (indicated by a gentle touch on the caudal tail with tweezers), and (2) the length of time it took for the fish to lose a “fin-pinch response” (defined as a pinch on the caudal fin with the tweezers, widely regarded as the fish “surgical plane” of anesthesia [25]). At this time, the fish was moved to a recovery tank with no tricaine, and the iPhone was positioned for recordings as described above. In this tank, the time to recover touch response was noted. The time to recover fin pinch response was not assessed because the fish had already been provoked. Videos were analyzed, and the rate of opercular movement was assessed for 10 s at several time points: (1) following the loss of touch response; (2) following the loss of fin-pinch response; (3) introduction into the recovery tank, and (4) 10 s prior to recovery of touch response as determined by the video recording. A timeline of this experiment is outlined in Appendix A. This opercular movement rate was extrapolated to determine opercular beats per minute. 

### 2.8. Heart Rate Analysis

Heart rate was assessed in WT Tübingen larvae, *chd7*^−*/*−^ larvae, and morpholino-injected larvae at 3 dpf. Genotype identity was masked to the researcher assessing heart rate to ensure the evaluation remained unbiased. Larvae were transferred one by one to a well of a 96-well plate in 2.5% tricaine to minimize movement for the duration of the analysis. Heart rate was assessed by observing the larva under a dissecting stereomicroscope and manually counting beats for 15 s, with at least 10 larvae/group. This experiment was completed 3 times. The same procedure was applied to the 3 dpf larvae from crosses of *chd7^+/^*^−^ fish injected with either *phox2ba-sfGFP* or *TagRFP* mRNAs, except that we measured the heart rates for 30 s and used the alternative genotyping method (see above).

### 2.9. Statistical Analysis

Statistical analyses and graphing were completed in GraphPad Prism (Version 8.1.1(224)). WISH categorical data were assessed with a Chi-squared test, and all other data were assessed using a one-way ANOVA with a Tukey post-test. For this study, a significance threshold was set to *p* = 0.05. 

## 3. Results

### 3.1. phox2ba Expression Is Reduced in chd7 Mutants 

We sought to determine the expression of the *phox2b* gene in zebrafish due to the role of this gene in CCHS and the importance of both *PHOX2B* and *CHD7* in normal neural crest development. As a result of full genome duplication events [26], there are two *PHOX2B* homologs in the zebrafish: *phox2ba* and *phox2bb.* We employed whole-mount in situ hybridization (WISH) to qualitatively assess both the spatial and temporal expression of these homologs. Tübingen larvae (WT) were used as controls. WISH was completed for both *phox2ba* and *phox2bb* and *sox10*, another neural crest gene. *Sox10* was previously shown to be regulated by Phox2b in a mouse model [2,5]. Larvae all displayed similar expression patterns for *phox2bb* and *sox10*, regardless of genotype (Appendix A). We then examined *phox2ba* expression patterns in WT embryos and all embryos produced from the heterozygous incross (expected to include *chd7^+/+^*, *chd7^+/^*^−^, and *chd7*^−*/*−^ larva; Figure 1A). We observed a subset of larvae with low staining levels, some with high staining levels, but the majority with moderate staining. The anticipated difference between the *phox2ba* expression patterns between the WT and the *chd7^+/^*^−^ led us to examine the correlation between gene expression and genotype further. 

To directly determine if *phox2ba* staining intensity correlated with genotype, we simultaneously performed in situ hybridizations of the larval heads (as *phox2ba* expression is restricted to the head at this time point) and genotyping of the caudal portion of the corresponding larvae. We performed this experiment in larvae collected from pooled clutches of a heterozygote cross (expected to include *chd7^+/+^*, *chd7^+/^*^−^, and *chd7*^−*/*−^ larvae) and WT embryos to increase the confidence of the WT data. We performed WISH and scored them into categories of low staining (score = 1), moderate staining (score = 2), or high staining (score = 3) (Figure 1A). The results of the scoring procedure identified an increased proportion of “low” staining scores in the *chd7*^−*/*−^ mutant group of larvae (Figure 1B). 

### 3.2. Morpholino-Mediated Knockdown of phox2ba Mimics the Decreased Heart Rates Observed in chd7^−/−^ Larvae

To determine if *phox2b* genes influence the autonomic nervous system function in zebrafish as PHOX2B does in humans, we transiently knocked down the expression of these genes using morpholinos (MOs). To examine the effects of each protein, we designed translation-blocking morpholinos for both *phox2ba* and *phox2bb*. Since MOs are only effective for approximately 72 h, we sought to define a larval phenotype that could be detected in both the *chd7* mutants and the *phox2b* morphants. When examining *chd7*^−*/*−^ mutant larvae, we determined that, in addition to many of the phenotypic developmental abnormalities seen in our previous studies (enlarged heart, smaller eyes, and a curved spine [19]), we also noted a decreased basal/resting heart rate (average of 168 vs. 182 beats per minute) at 3 days post-fertilization (dpf) (Figure 2A). With this larval phenotype established, we sought to determine if *phox2b* morphants had a similar phenotype. We injected the *phox2ba* and *phox2bb* morpholinos at 0.1, 0.25 or 0.5 mM to determine the optimal concentration. Even at the lowest doses, the knockdown of *phox2bb* resulted in generalized toxicity that impaired heart rate assessment. By contrast, injection of 0.1 mM *phox2ba* MO significantly reduced the HR in larvae, compared to control MO-injected larvae (Figure 2B). Higher doses of *phox2ba* MO resulted in mortality. 

We then attempted to rescue the heart rate phenotypes of *chd7*^−*/*−^ mutant larvae with over-expression of *phox2ba-sfGFP* mRNA (Appendix A). Upon injection, we observed robust expression of the *phox2ba-sfGFP* fusion at 28 hpf (Appendix A) in the embryos from the chd7^+/−^ cross, it did not persist until 3 dpf. Nevertheless, given the possible role of early Phox2ba-driven effects, we recorded heart rates of *phox2ba-sfGFP* mRNA-injected and control (*TagRFP* mRNA) larvae at 3 dpf in an unbiased manner, genotyped them, and determined that Phox2ba-sfGFP over-expression did not increase the heart rates of *chd7*^−*/*−^ mutant larvae or alter heart rates of larvae with other genotypes (Appendix A). This finding suggests that stable transgenic approaches are required to clarify the role of *phox2ba* in *chd7* mutant zebrafish.

### 3.3. Chd7 Mutant Adults Exhibit Differential Response to Anesthesia and Opercular Rates

Adult zebrafish (previously genotyped as *chd7^+/+^*, *chd7*^−*/+*^, or *chd7*^−*/*−^) at approximately 3 months of age were examined for both behavioral and respiratory responses to anesthesia. Only a small number of *chd7*^−*/*−^ fish were identified in adult stocks of the mutant line, suggesting that most *chd7*^−*/*−^ fish died in the larval stage. Briefly, fish were put into experimental tanks containing tricaine and videotaped. While the time to loss of touch response was higher in *chd7*^−*/*−^ fish than both *chd7^+/+^* and *chd7^+/^*^−^ fish, there was a high degree of variability (Figure 3A). However, neither time to loss of fin-pinch response nor time to recovery of touch response were significantly different between genotypes (Figure 3B,C). 

We next looked at opercular beat rate as a measure of respiratory rate, as reported by others [27]. We recorded opercular rates throughout anesthesia and observed that opercular rates were not significantly different among the three groups of fish at the point of loss of touch response. However, strikingly, later in the experimental timeline, when the surgical plane of anesthesia was achieved (measured by lack of a fin pinch response [25]), and during the initial recovery period, the opercular beat rate in the *chd7^−/−^* fish was significantly (*p* = 0.0009) higher in comparison with the *chd7^+/−^* and *chd7^+/+^* fish groups (Figure 4). 

## 4. Discussion

CHARGE syndrome and CCHS share several similarities. Each of these monogenetic syndromes is thought to arise due to mutations in a single gene (*CHD7* and *PHOX2B*, respectively) that regulates neural crest development [28,29]. Each syndrome is associated with debilitating defects in the control of the ANS. In CCHS, patients cannot detect the low oxygen levels in their blood, which often leads to hypoventilation while sleeping, which can result in death [30]. In individuals with CHARGE syndrome, autonomic dysfunction presents in various ways, with gastrointestinal motility issues [10], heart defects, and dysregulated respiration, especially after anesthetic [6]. Additionally, the hypoventilation seen in CCHS is similar to clinical observations of CHARGE patients recovering from anesthesia. It has been reported that these patients exhibit decreased respiratory rates and abnormal breathing patterns during recovery [6]. In several cases, patients have unidentified postoperative airway events that may resemble hypoventilation. Perhaps unsurprisingly, patients with CCHS also are vulnerable to adverse events following anesthesia [10]. Interestingly, a case report describes a neonate with severe mixed apneas requiring respiratory support that was markedly improved by a tracheostomy. Further genetic testing revealed that this infant had CHARGE syndrome [31]. Considering these clinical similarities, the current study utilized a previously established CRISPR-based *chd7* knockout zebrafish model of CHARGE syndrome to assess the potential overlap between these two neurocristopathies. 

WISH demonstrated that *phox2ba*, *phox2bb*, and *sox10* were all expressed in zebrafish larvae. There were no differences in *phox2bb* or *sox10* anatomic expression or intensity in the progeny resulting from an incross of *chd7^+/−^* fish (expected to include *chd7^+/+^, chd7^+/−^*, and *chd7^−/−^* larvae) and WT larvae (Appendix A). In contrast, *phox2ba* expression differed in our *chd7* knockout fish, with a higher percentage of fish showing reduced expression (Figure 1). The differential spatial expression of *phox2ba* in our zebrafish model of CHARGE syndrome suggests that, in addition to mutations in *CHD7*, there may be differences in the expression of other genes involved in regulating neural crest development. To our knowledge, this is the first report of a potential association between PHOX2B and CHD7. While there are other valuable zebrafish models of CHARGE syndrome [16,17], we believe this is the first CHARGE model in zebrafish in which behavioral assessment was conducted.

Next, we set out to investigate the physiological effects of *phox2ba* reduction experimentally using transient gene knockdown. *phox2ba* knockdown reduced larval heart rates, similar to what is seen in *chd7^−^^/−^* larvae at baseline (Figure 2). This suggests that *phox2ba* and *chd7* may play similar roles in the autonomic control of larval heart rate [32]. The lack of effects from Phox2ba-sfGFP fusion over-expression on the heart rates of *chd7^−^^/−^* larvae could be explained either by the limited persistence of Phox2ba-sfGFP fusion after mRNA injection or, more likely, by the more complex program involved in heart rate regulation that is affected by *chd7* loss. 

Behavioral experiments in adult fish demonstrated that *chd7^−^^/−^* fish had differential responses to anesthetic compared with WT fish (Figure 3 and Figure 4). Specifically, the *chd7^−^^/−^* fish took longer to lose the touch response (Figure 3A). While there was a high degree of variability within this experimental group, this is in keeping with the significant variability in the clinical features of individuals with CHARGE syndrome. Further, while the *chd7^−^^/−^* fish reached the surgical plane of anesthesia, their respiratory rates did not decrease to the same degree as their WT counterparts (Figure 4). This fits with clinical observations of CHARGE patients requiring higher anesthetic doses to become anesthetized [7]. It is worth noting that the loss of fin-pinch response is used experimentally in fish as a benchmark for the surgical plane of anesthesia [25]. Thus, while the *chd7^−/−^* did reach the surgical plane of anesthesia, their respiratory rate was significantly elevated in comparison with their *chd7^−/+^* and *chd7^+/+^* counterparts, suggesting they may require higher doses of anesthetic to reach the same level of sedation. We used the zebrafish muscle relaxant tricaine as a surrogate for the inhalant anesthetics to which patients with CHARGE syndrome are exposed. While the pharmacologic class of tricaine is that of a local anesthetic, it has systemic effects in fish, as it is absorbed via their gills and integument [33]. 

Our zebrafish model has provided new insights into the biology underlying how CHARGE patients experience anesthesia. Experimentally, *chd7^−/−^* fish respond differently to anesthesia by taking longer to become anesthetized and having significantly higher respiratory rates while within the surgical plane of anesthesia. This suggests that this zebrafish model could be used as a preclinical platform to assess different classes of anesthetics and to optimize anesthetic choice for individuals with CHARGE syndrome. CHARGE syndrome is currently treated symptomatically, with interventions that usually involve procedures and surgeries requiring anesthesia. Our results support the concept of performing multiple surgeries under one anesthetic, thereby reducing risks associated with repeated anesthesia [6]. By investigating the roles of *CHD7* and *PHOX2B* in CHARGE, we have furthered our understanding of disease pathogenesis, with the overall goal of making anesthesia a safer process in individuals with CHARGE syndrome. 

## Figures and Tables

**Figure 1 genes-14-01086-f001:**
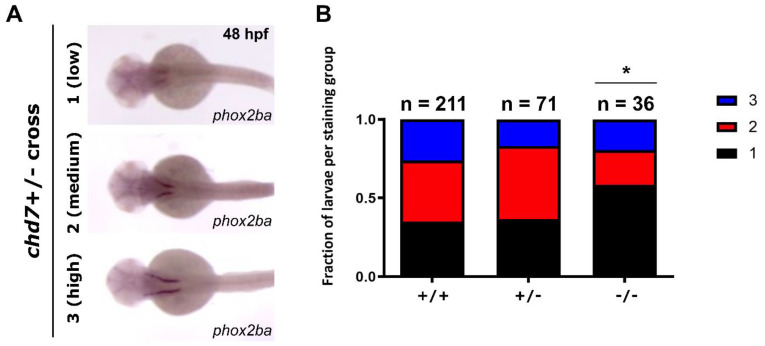
*phox2ba* expression is reduced in *chd7* mutants compared to sibling embryos. (**A**) Representative images of *chd7*^+/−^ cross embryos stained for *phox2ba* at 48 h post−fertilization (hpf) with scored phenotypes. Note that there are variable levels of expression ranging from “low” = 1, “medium” = 2, and “high” = 3. (**B**) Phenotype fractions bar graph of *phox2ba* staining in (**A**). Staining patterns were quantified by scoring based on intensity. The numbers of embryos scored for each genotype are indicated above the graphs. Note the increased proportion of embryos with weak staining patterns in the *chd7*^−*/*−^ group. (Chi-square *p* = 0.0363 (indicated by ‘*’ above the mutant bar).

**Figure 2 genes-14-01086-f002:**
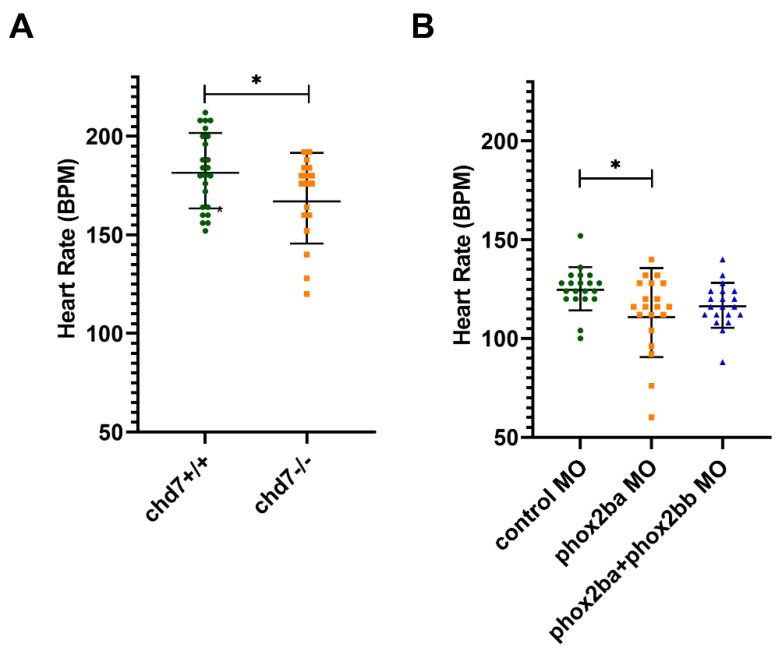
*chd7*^−*/*−^ and *phox2ba* morpholino (MO) −injected larvae have lower heart rates than their WT or uninjected counterparts. (**A**) Heart rates were counted in zebrafish larvae under anesthesia at 72 h post−fertilization (hpf). * *p* ≤ 0.05, as per the two-tailed unpaired *t*−test, error bars represent SD, and each point represents an individual larva, green represents *chd7^+/+^*, orange represents *chd7^+/+^* (+/+ n = 24, −/− n = 18). (**B**) Zebrafish Casper embryos were injected with morpholinos (MOs) at the one−cell stage to knock down RNA expression, and heart rates were counted under anesthesia at 72 hpf. Embryos injected with *phox2ba* MO had significantly lower heart rates than control MO−injected fish. * *p* < 0.05, as per one−way ANOVA with a Tukey post−test, error bars represent SD, each point represents an individual larva, green represents control MO, orange represents *phox2ba* MO, blue represents *phox2bb* MO n ≥ 9 larvae/group.

**Figure 3 genes-14-01086-f003:**
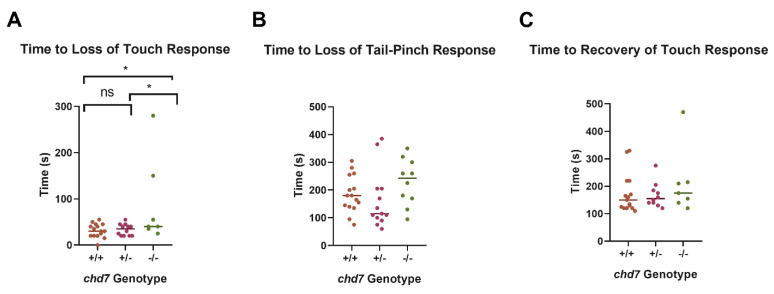
*chd7−/−* fish take longer to lose touch response under anesthetic. Anesthesia was assessed by the time it took adult zebrafish (previously genotyped as *chd7^+/+^* [brown], *chd7^+/−^* [purple] or *chd7^−/−^* [green]) to lose their response to touch (**A**) and response to tail−pinch (**B**). Recovery was measured as the time it took the fish to regain the touch response (**C**). * *p* < 0.05, as per one−way ANOVA, with a Tukey post−test.

**Figure 4 genes-14-01086-f004:**
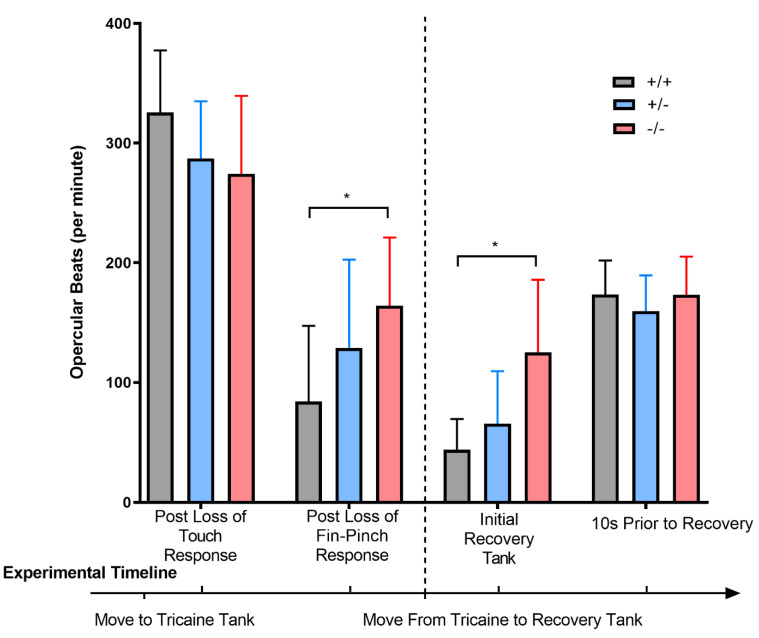
*chd7^−/−^* fish have higher respiratory rates under anesthesia than their *chd7^+/+^* counterparts. Adult fish were anesthetized, and respiratory rates (opercular movements) were recorded at the time fish lost their response to touch (time point 1− post loss of touch response) and response to fin−pinch (time point 2− post loss of fin pinch response). Fish were then moved to a recovery tank, and respiratory rates were assessed both upon entering the recovery tank (time point 3− initial recovery tank) and just before the fish recovered the ability to swim (time point 4–10 s prior to recovery). * *p* < 0.05, as per one-way ANOVA, with a Tukey post-test. Error bars represent standard deviation (SD). n = 7 for *chd7^−/−^* and n = 13 for the other two groups.

## Data Availability

The raw data presented in this study are available on request from the corresponding author. The data are not publicly available for privacy reasons.

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
