# Peer review of "phox2ba: The Potential Genetic Link behind the Overlap in the Symptomatology between CHARGE and Central Congenital Hypoventilation Syndromes"

_genes, 2023, doi:10.3390/genes14051086_

Round 1

Reviewer 1 Report

In this study, MacLean et al. investigated the expression of phox2b genes and anesthetic recovery behaviour in a zebrafish model of CHARGE syndrome. Overall, the paper is well written but the authors should dampen the conclusions on the link between PHOX2B and CHD7 they seem more speculative than based on the presented data.

The authors report that phox2ba expression is reduced in chd7 mutants compared to sibling embryos. Yet the staining patterns and variations including in WT are intriguing. Could developmental delay in chd7-/- be the reason for  the increase in the number of low staining at 48 hpf?  

The genetic link/functional link between phox2ba and chd7, as claimed by the authors based on their observations, remains weak and speculative. To strengthen their claim, it is important to (i) check the level of phox2ba in chd7-/- to see if it is downregulated and (ii) perform overexpression studies to see whether phox2ba can rescue/ameliorate phenotypes in chd7-/-.  Or modify the conclusion of the paper.

Minor comments:

- CHARGE syndrome has been modelled in zebrafish by several groups and these studies should be cited to show the validity of the model for the disease.

- Is the control MO a mismatched or a standard control? Ideally a mismatched MO and/or splicing MO should be used. 

- Figure 1B – please correct -\+ to +/- in the figure labeling.

 “Jason N Berman3 and 4 #” Please correct the author list - remove "and".

Reviewer 2 Report

The manuscript submitted by MacLean and colleagues is of interest for several reasons. First, they establish that chd7 homozygous mutants have less expression of phox2ba and that both chd7 homozygous mutants and phox2ba morpholino injected larvae have lower heart rate suggesting a link between the two genes. In addition, they provide evidence that chd7 homozygous mutants have higher respiratory rates under anaesthesia.

CHARGE is clearly an heterogeneous syndrome so it is interesting that it may expand into the CCHS syndrome. This kind of information is important for disease stratification.

The authors use careful wording to highlight this is a "first report of a potential association between PHOX2B and CHD7" but many questions remain open.

Major issues

Figure 1 suggests chd7 drives the expression of phox2b. A rescue experiment where phox2b is overexpressed together with functional data or more direct evidence of a link between the two genes would have been valuable although this is clearly a major undertaking and possibly beyond the scope of this paper.

Minor issue

the presence of two genes for phox2b should be highlighted in the introduction

The meaning of the sentence between line 302 to 304 is not very clear

Line 191 to 194. Authors should indicate in Fig1S phox2bb and sox10 expression in homozygous instead of heterozygous since the expression profile of phox2ba is normal in hets

I assume heterozygous incross means het-by-het cross using isogenic animals.

Figure 1A should be labelled with the name of the probe ie phox2ba. Figure 1B y axis should stop at 1.

Line 168 169, editors may want to have figures referred in order of appearance. Check if material and methods is going at the end or beginning and change name of figure 4 accordingly.

chd7 mutations are lethal given the number of heterozygous and homozygous mutants in figure 1 compared to expected ratios if this is a het by het cross. This suggests that both hets and homozygous are equally affected (you'd expect twice as much hets than homs, which is the case here 71 vs 36) in terms of viability. So perhaps, this explains why animals that made it to late embryos and adulthood are almost normal and only a few display a clear phenotype, such as the couple of outliers in figure 3A. Phenotypic variability is a hallmark of CHARGE as mentioned by the authors even within isogenic models. Partial penetrance of certain phenotypes in certain models is a real puzzle to explain.

It is unclear if the effect of phox2bb MO is toxicity or mirrors the lethality phenotype, what is the authors view on this ?

Figure 2 seem to combine data from different sources. It should be made clear if part of this is published elsewhere.

Reviewer 3 Report

The manuscript submitted by MacLean et al describes a putative connection between the CHARGE and Central congenital hypoventilation syndromes. According their observations, the frequent adverse respiratory post-anesthesia events detected in CHARGE patients could be the result of defects in PHOX2B expression.

In a zebrafish model of CHARGE (by CRISPR/Cas knockdown) they describe lower levels of phox2b genes (in fact only phox2ba, not phox2bb). In Morpholino experiments knocking-down phox2ba they detected decreased heart rate as in chd7-/- larvae. Then the authors describe how chd7-/- respond differently to anesthesia (Tricaine) with respect to control wild type larvae.

Although it is true that CHARGE syndrome and CCHS share several similarities, the evidence provided by the authors is not conclusive to support their conclusions.

Major concerns

1-Confirmation of phox2ba knock-down by Morpholino injection is required. qPCR, WISH or western blotting detecting the phox2ba decreased expression is essential to confirm the CCHS model. Rescue experiment would be of value too.

2-In order to associate CHARGE respiratory manifestations to phox2ba knock-down it is essential to perform rescue experiments of CHD7-/- embryos with phox2ba mRNA and check the heart rate. Also, opercular beats response could be interesting to analyze. Otherwise, the title of the manuscript should be changed in order to moderate the association.

Minor concerns

1-Verification by qPCR of the expression of sox10, phox2ba and phox2bb in chd7+/+, chd7+/- and chd7-/-. To this reviewer phox2bb expression seems lowered in figure S1a. Also sox10 expression.

2-Figure 2 should be improved. Both graphs should have the same size. Also the same scale in the y axis. Why controls in figures 2 A and 2B exhibit different heart rate?

3-Figure 4. The experimental timeline should be more specific and clearer.

4-References #22 and #24 are the same.

Round 2

Reviewer 1 Report

The authors have addressed the concerns raised very well. The manuscript has considerably improved and is suitable for publication.

Author Response

We thank the reviewer for their valuable input. 

Reviewer 3 Report

The manuscript has been improved.

Author Response

We thank the reviewer for their input.